# Chemoprevention of Colorectal Cancer—With Emphasis on Low-Dose Aspirin and Anticoagulants

**DOI:** 10.3390/ph18060811

**Published:** 2025-05-28

**Authors:** Arnar Snaer Agustsson, Einar Stefan Bjornsson

**Affiliations:** 1Faculty of Medicine, Landspitali, University Hospital of Iceland, 101 Reykjavik, Iceland; arnarsn93@gmail.com; 2Faculty of Medicine, University of Iceland, 101 Reykjavik, Iceland

**Keywords:** colorectal cancer, aspirin, oral anticoagulation, metformin, chemoprevention, pharmacoepidemiology

## Abstract

**Background and Aims:** Colorectal cancer (CRC) remains the third most common cancer worldwide and a leading cause of cancer-related death. Chemoprevention through widely used pharmaceutical agents has garnered increasing interest due to its potential cost-effectiveness and accessibility. This review summarizes current evidence from observational studies, randomized controlled trials, and meta-analyses on the association between commonly prescribed medications and CRC incidence and survival, with particular emphasis on low-dose aspirin and oral anticoagulants (OACs). **Scope:** Aspirin is the most extensively studied agent, with substantial evidence supporting its protective effect on CRC-specific survival, particularly in long-term users, those with COX-2 overexpression, or PIK3CA mutations. OACs have recently gained attention due to their association with increased gastrointestinal bleeding, which may facilitate earlier CRC detection. While emerging evidence suggests a possible survival benefit through this mechanism, data remain heterogeneous and affected by methodological challenges such as lead-time bias. Metformin is associated with improved CRC outcomes, primarily in patients with type 2 diabetes, though its direct anti-tumor potential remains under investigation. Corticosteroids, statins, and beta-blockers have both limited and inconclusive evidence. Finally, recent studies on vitamin D, calcium, and folic acid suggest inconsistent associations, often confounded by lifestyle factors or underlying comorbidities. **Conclusions:** While promising, chemoprevention strategies require further validation in well-designed, mechanistically informed studies that account for confounding variables, treatment duration, and tumor biology. Personalized prevention—guided by genetic, molecular, and clinical risk factors—represents a promising path forward.

## 1. Introduction

Colorectal cancer (CRC) is the third most common cancer worldwide, with approximately two million new cases diagnosed annually, and it remains the second leading cause of cancer-related death globally [1]. CRC incidence mirrors socioeconomic development; while incidence rates have stabilized or decreased in developed nations, developing countries are facing a rapid increase in CRC burden [1]. As CRC continues to pose a growing global health challenge, particularly in low- and middle-income countries, there is increasing interest in preventive strategies that are accessible, cost-effective, and scalable. One such measure has been chemoprevention, as certain medications could serve as a cost-effective and accessible strategy to reduce CRC incidence or mortality. This is of importance given both the rising incidence of CRC in developing countries and the increasing use of common medications, especially aspirin and oral anticoagulants (OACs) [2,3].

However, studies examining medication use and cancer-related survival face notable methodological challenges, including immortal time bias, lead-time bias, the healthy user effect, and confounding by indication. Accurate assessment of medication exposure and disentangling mechanisms—whether affecting cancer progression, underlying risk factors, or detection—remains an ongoing challenge. Despite these complexities, high-quality pharmacoepidemiologic research that employs rigorous study designs and advanced statistical methods continues to provide valuable insights. In this review, we summarize current evidence on the association between CRC outcomes and the use of aspirin, OACs, metformin, and corticosteroids, and briefly discuss emerging data on beta-blockers and statins.

## 2. Results

### 2.1. Aspirin and Colorectal Cancer

Aspirin (or acetylsalicylic acid) is an irreversible cyclooxygenase (COX) enzyme inhibitor widely used for the secondary prevention of myocardial infarction or stroke [4]. In an original publication from 1988 in Cancer Research, aspirin users developed CRC less often compared to non-users, suggesting that aspirin could affect the incidence of CRC [5].

Aspirin’s inhibition of COX-2 likely reduces prostaglandin-mediated tumor progression, supporting its role as a chemopreventive agent. Inspired by this finding, numerous studies have since been conducted on the effects of aspirin on CRC-specific survival, including large, long-term observational cohort studies, randomized controlled trials (RCTs), and meta-analyses. Table 1 summarizes the estimated effects of aspirin on CRC-specific survival and aspirin dose for notable and highly cited studies. This scientific body demonstrated that aspirin use might greatly reduce CRC-specific mortality by around 20–30%.

In 2016, the U.S. Preventive Services Task Force (USPSTF) recommended the use of low-dose aspirin for primary prevention of CRC in patients aged 50–59 years old, given that they would be willing to take low-dose aspirin daily for at least 10 years and were not at increased risk of bleeding events [27]. They also recommended that patients 60–69 years old should make an individualized decision of taking daily low-dose aspirin for primary CRC prevention, granted they were not at increased risk of bleeding [27]. However, no recommendation was given for 70–79-year-old patients, primarily due to a lack of evidence, and the fact that primary prevention occurred after consistent long-term use of aspirin, which is less applicable in the older cohort.

Unexpectedly, when examining mortality in the elderly, aspirin use was associated with increased CRC incidence and CRC-specific mortality (HR = 1.77; 95% CI: 1.02–3.06) [19]. Subsequently, this was studied in more detail, and increased mortality was found for all solid cancers [28]. This contrasts with the evidence presented in Table 1, which includes patients over 70, so these results should be interpreted carefully. Additionally, a recent study found that patients over 70 had protective effects only if they had initiated aspirin therapy before age 70 [29]. Thus, different studies have shown conflicting results. Therefore, there is a need for future studies on aspirin use and its association with both CRC incidence and CRC mortality in the group aged over 70, to determine whether aspirin use could be beneficial or harmful.

The beneficial effects of aspirin on CRC-survival have consistently been shown to take a few years to develop. This was most prominently observed in the Women’s Health Study conducted by Cook et al., a randomized controlled study (RCT) published in 2005, which did not find survival benefits when comparing aspirin use to placebo over a 10-year follow-up period [20]. However, in the 8-year post-trial follow-up, protective effects of aspirin use emerged with a sharp post-trial reduction of CRCs of 42% (HR = 0.58; 95% CI: 0.42–0.80, *p* < 0.001) [13]. This finding of delayed effects has been consistently demonstrated in multiple large studies, finding that >5 years of aspirin therapy effectively improves CRC survival [7,15,16,30,31].

However, studies examining aspirin use initiated after CRC diagnosis have also shown CRC-specific survival benefits in aspirin-naïve patients at CRC diagnosis [10,17,32,33], with some studies suggesting no survival benefits [34,35]. Additionally, in a study by Rothwell et al., aspirin use was associated with a lower risk of metastatic disease compared to non-users, especially in CRC (HR = 0.26; 95% CI: 0.11–0.57, *p* = 0.0008) [36]. These results could indicate that aspirin could delay or hinder the metastatic development of CRC. This is further supported by studies finding that the COX-2 enzyme plays a key role in promoting tumor metastasis [37,38,39,40,41,42]. Therefore, this could be a crucial mechanism by which aspirin lowers CRC mortality, both in overall patients, as a second prevention for current CRC patients, and in aspirin-naïve patients. This is most likely just one key mechanism by which aspirin could mediate its beneficial effects, as significant differences in stage IV diseases between aspirin users and non-users have not been observed.

The discovery of aspirin’s chemopreventive effects on CRC mortality has led to numerous studies being conducted on the potential mechanism. Two mechanisms have been proposed: direct anti-tumor activity of aspirin or early detection through increased bleeding events.

Aspirin’s inhibition of the COX enzymes is often believed to be the primary mediator of the CRC survival benefits. COX-2 enzyme expression in CRC is involved in apoptosis, angiogenesis, and the invasiveness of the cancer [43]. Several studies have demonstrated that the COX-2 enzyme is overexpressed in most CRCs [44,45,46,47]. Furthermore, studies have shown explicit survival benefits in patients using aspirin and having CRCs with COX-2 overexpression or having a mutation in the phosphatidyl-inositol 3-kinase (PI3K) pathway, causing increased COX-2 signaling [14,17,47]. This was demonstrated by Liao et al., who found that aspirin use after CRC diagnosis significantly improved CRC-specific survival in patients with PIK3CA-mutated CRCs (HR = 0.18; 95% CI: 0.06–0.61, *p* < 0.001). This is important because it suggests that aspirin might affect only a part of the population diagnosed with CRC and thus might explain why some studies have found no effect (see Table 1). In the study by Liao et al., only 17% of CRCs had PIK3CA mutations [14], while a study by Chan et al. found 67% of tumors had COX-2 overexpression [47].

Aspirin’s effects on CRCs with the PIK3CA mutation are a hot research topic [48], with results from an RCT that examined disease-free survival in CRC patients diagnosed at stages II–III, finding a clear trend towards improved survival in aspirin users [49]. Additionally, the preliminary results from the ALASCCA trial, a multicenter Nordic RCT, demonstrated an over 50% reduction in CRC recurrence in aspirin users when examining patients with PIK3CA-mutated CRCs [50]. Further results from the ALASCCA trial and future RCTs examining the PIK3CA mutations in CRCs will illuminate the effects of aspirin use and, with further risk/benefit studies on aspirin use, could lead to aspirin being used as a component of adjuvant therapy in CRCs.

Further supporting aspirin’s anti-tumor effects, several studies have demonstrated a reduction in colonic polyps among high-risk individuals. Regular long-term use of aspirin has been shown to reduce the incidence of CRC [51,52,53,54]. The RCTs examining aspirin and colonic polyps have demonstrated aspirin’s protective effects, particularly in patients with prior CRCs or colonic polyps [55,56], but not in men with average risk of CRC [57]. This suggests that aspirin’s effects are not evenly distributed, as they may not carry overall protective effects for the population but may carry protective effects for individuals at high risk or with certain mutations.

One of the most important adverse effects of aspirin is the increased risk of bleeding, especially gastrointestinal bleeding (GIB) [58,59,60]. Since lower GIB events are the most common presentation of CRC [61,62,63], this could cause aspirin users to have increased detection through increased bleeding events. However, it would be expected that, on average, aspirin users would have been diagnosed at earlier stages compared to non-users in the observational studies in Table 1. To address this gap in the literature, further studies are required to assess the relationship between aspirin use, GIB events, and CRC, as it is unlikely that there would be a consistent survival benefit due to early diagnosis without staging being different in aspirin users compared to non-users.

Lastly, aspirin use has been associated with protective effects in patients with Lynch syndrome [64,65]. Lynch syndrome, caused by germline mutations of DNA mismatch-repair genes (MMR), accounts for approximately 5% of all CRC, but carriers of these mutations have a lifetime risk of around 50% of CRC [66]. The CAPP2 study was a multicenter RCT that compared 600 mg of aspirin to a placebo in patients with Lynch syndrome, and aspirin reduced CRC risk substantially (HR = 0.65; 95% CI: 0.43–0.97, *p* = 0.035) [67].

All this evidence supports the recommendation that for patients without an increased bleeding risk at baseline and able to take low-dose aspirin for 10 years or longer, aspirin should be recommended to prevent CRC. Use of aspirin after CRC treatment is promising, but should most likely be personalized and based on tumor markers. Finally, for patients with increased CRC risk at baseline, aspirin as chemoprevention for CRC should be seriously considered.

### 2.2. Oral Anticoagulation and Colorectal Cancer

The use of oral anticoagulation (OAC), including vitamin-K antagonists (VKA) or direct oral anticoagulants (DOACS; including rivaroxaban, apixaban, dabigatran, and edoxaban), has been increasing in recent decades [68,69]. Their most common serious adverse effects are the occurrence of GIB events, which has led to the hypothesis that their use might lead to early detection of tumors in the gastrointestinal tract. Since GIB events are the most common presentation of CRC [61,62], there has been great interest in whether OAC use could cause early detection of CRC. Furthermore, OAC users experienced gastrointestinal bleeding from CRC at higher rates than non-users [70], supporting the hypothesis that these medications may help unmask otherwise silent tumors and facilitate earlier diagnosis.

Multiple observational cohort studies have demonstrated a higher incidence of CRC diagnosis among OAC users than non-users, supporting the hypothesis that anticoagulation-induced gastrointestinal bleeding may facilitate earlier detection of colorectal malignancies, as summarized in Table 2. As shown in Table 2, these studies exhibit variability and employ different statistical measures, making direct comparison challenging. However, they all either compare CRC incidence, CRC detection, or CRC-specific mortality, and they all point to the conclusion that OAC use could improve CRC survival by facilitating earlier detection.

To our knowledge, only one study has examined CRC mortality as an endpoint, finding a clear trend towards improved survival in VKA users [75]. That study is an administrative database study and did not evaluate variables such as GIB events. Additionally, data availability across subgroups was inconsistent. Therefore, there is a need for a study evaluating OAC use, GIB events, and both CRC staging and CRC mortality to determine if there is an association of OAC use, bleeding events, and improved cancer survival. One of the challenges of such a study would be to determine whether an observed improvement is an actual improvement or could be due to lead-time bias, resulting from an earlier discovery of CRC, which would falsely contribute to increased survival.

The Danish population-based and primary care study on GIB events found that patients on OACs had more GIB events caused by CRC than non-users, irrespective of age [71]. This supports the hypothesis that OAC use may facilitate earlier detection of CRC through bleeding events, potentially leading to earlier diagnosis and improved outcomes.

An alternative mechanism has been proposed, since VKAs might have direct anti-tumor effects by inhibiting tumor cell migration and invasiveness, irrespective of their effects on coagulation, through inhibiting the Axl tyrosine kinase receptor [77]. Due to that anti-tumor effect, a large population-based cohort study from Norway found warfarin use to be associated with a lower incidence of CRC when examining warfarin users due to atrial fibrillation, but not when examining overall warfarin users [73]. This, with the other studies in Table 2, supports a GIB mechanism of increased CRC incidence, since similar results are found for studies of DOACs and when examining VKAs and DOACs together. These findings further underscore the need for future studies on the association between OAC use and CRC survival to include important clinical variables, such as GIB events and histopathology, in an attempt to elucidate the underlying mechanism.

Notably, most included studies did not provide stratification by specific OAC dose, limiting the ability to evaluate dose-response relationships. For example, only Clements et al. reported DOAC dose [74], while all other studies in Table 2 reported OAC use as either “yes” or “no”. This represents an important gap for future pharmacoepidemiologic research. An additional gap is that OAC users are most commonly over 65 years old and are more often males [78,79,80], potentially limiting their external validation to female patients and other subgroups.

With the increasing use of OACs, and particularly DOACs, there is a clear need to determine if their use could lead to early CRC detection with improved survival. If a protective association is found, that could suggest that OAC users should undergo systematic follow-up and potentially justify earlier screening or shorter time intervals between screenings.

### 2.3. Metformin and Colorectal Cancer

Most studies have indicated that metformin use is associated with lower CRC risk. Metformin use has been associated with decreased CRC risk, finding metformin to both lower the incidence of CRC (RR = 0.76; 95% CI: 0.69–0.84, *p*  <  0.001) and increase the CRC-specific survival (HR = 0.66; 95% CI: 0.59–0.74, *p*  <  0.001) [81]. Three meta-analyses suggest the same CRC-incidence lowering effects in metformin users compared to non-users [82,83,84], and a meta-analysis from Mei et al. found a 34% decrease in CRC-specific mortality (HR = 0.66; 95% CI: 0.50–0.87) [85]. Meanwhile, two large, quality observational studies have not demonstrated that metformin use lowers the CRC incidence [86] or CRC mortality [87]. However, observational studies assessing metformin dose have found an association between lower CRC risk and higher metformin dose [88,89].

Another added complexity to the metformin literature is the proposed effects of metformin. Metformin is used to treat type 2 diabetes (T2D), and two meta-analyses have demonstrated that patients with T2D have a higher risk of CRC incidence [90,91] and CRC mortality [90]. This has been further demonstrated by the fact that elevated hemoglobin A1c (HbA1c) is associated with worse short-term outcomes, such as more aggressive cancer and a higher rate of post-operative complications, perhaps due to impaired metabolic control in these patients [92,93]. Additionally, in a meta-analysis, elevated HbA1c was associated with higher incidence and mortality from CRC [94].

In contrast, a large, well-conducted population-based study from Sweden found no association between T2D and increased CRC risk [95]. However, other studies have found that patients with T2D and CRC have an increased risk of CRC mortality compared to CRC patients without T2D [96,97]. This inconsistency underlines the further need for large studies assessing HbA1c levels and the CRC risk.

Some studies have indicated that the potential mechanism of metformin’s beneficial effects is due to a reduction in tumor cell growth by the AMPK pathway [98,99,100]. This suggests that metformin’s effects are not directly connected to its effects on patients with T2D. However, no studies have been conducted on metformin use outside of patients with T2D, and such studies are unlikely to be carried out because the potential benefit is unclear and metformin use has potential adverse effects. Additionally, separating the potential beneficial effects of metformin from the increased risk in patients with T2D is challenging.

Even though the hypothesis of potential direct anti-tumor effects of metformin is intriguing, its clinical benefit may largely reflect the mitigation of diabetes-related oncogenic risk. Large, high-quality studies examining patients with T2D and assessing the severity of T2D using end-organ damage or HbA1c and the metformin dose would shed further light on the association of T2D, metformin use, and CRC incidence and CRC mortality.

Therefore, while metformin has been shown to decrease CRC incidence and mortality, the benefit is as yet limited only to patients with T2D, and there is no convincing evidence to suggest that metformin is associated with a decreased burden of CRC for other populations. Future studies should focus on metformin doses, time to effect, and the severity of T2D in relation to CRC incidence or CRC mortality as endpoints. Additionally, investigating a potential dose–response relationship or a link between the severity of diabetes and CRC risk would be beneficial.

### 2.4. Corticosteroids and Colorectal Cancer

Corticosteroids have complex and multifactorial anti-inflammatory effects that may influence CRC progression, including NF-kappa beta transcription inhibition, pro-inflammatory cytokine inhibition, activation of anti-inflammatory genes, and T-cell modulation to suppress inflammatory response [101,102,103,104]. Since chronic inflammation plays a critical role in cancer development [105,106], this could be an important association in corticosteroid users.

Since chronic inflammation plays a critical role in cancer development, it is unsurprising to see patients with Ulcerative Colitis (UC) and Crohn’s disease (CD) having an increased risk of CRC [107,108]. Corticosteroids have been shown to reduce the risk of CRC in UC and CD [109]. However, whether corticosteroids only mitigate the increasing risk in these populations or provide decreased risk to a broader population has not been determined. Prolonged corticosteroid use in the general population has been associated with increased skin and bladder cancers [110,111,112,113].

Corticosteroids suppress inflammation by inducing a sublineage of T-cells, T-regulatory cells, that actively suppress other T-cells [103,104]. Studies have found that increased T-regulatory cell presence in the microenvironment of the tumor is associated with improved survival [114,115], likely by dampening harmful inflammation. However, the cancer could also use these T-regulatory cells to evade the immune system; a study found that all patients with CRC recurrence had the same genetically modified T-regulatory cells, accounting for a third of all T-regulatory cells in the study [116]. Therefore, there is some evidence to suggest that corticosteroids might have either protective effects, by reducing harmful inflammation, or negative effects by increasing tumor evasion. 

Two population-based cohort studies have been conducted on the effects of prior corticosteroid use and CRC incidence without finding any clear association [117,118]. However, a recent meta-analysis linked corticosteroid therapy with higher mortality and increased disease progression in solid cancers, including CRCs [119]. The limitations of those studies are that they only examined steroid use after cancer diagnosis and therefore could be more of a marker of disease severity rather than a mechanistic causation of worse prognosis.

To our knowledge, no study has been conducted on corticosteroids and CRC-survival, and with the potentially complex and multifactorial effects of corticosteroids on CRC progression, this remains a promising area for future research. The challenge of these studies will be to identify a dose–response relationship, as corticosteroid use varies greatly based on the indication for treatment and, if a clear association exists, to elucidate the potential mechanism by which corticosteroid use could increase or decrease the risk of CRC.

### 2.5. Statins, Beta-Blockers, and Colorectal Cancer

Evidence on the potential beneficial effects on CRC risk of statins (or 3-hydroxy-3-methylglutaryl coenzyme A (HMG-CoA) reductase) has been debated. Statins’ inhibition of cholesterol synthesis could cause a decrease in tumor growth, a basis for their potential anti-tumor effects [120]. An RCT demonstrated protective effects against the recurrence of CRC in patients with prior CRC diagnosis [121]. Two meta-analyses have demonstrated a modest reduction in CRC risk in cohort studies, but not in RCTs [122,123]. Another meta-analysis demonstrated CRC risk benefits, both for overall and CRC-specific mortality, and both for pre-diagnosis and post-diagnosis statin use [124]. The CRC-specific mortality reduction of statins was demonstrated again in two meta-analyses [125,126]. This inconsistent data, heterogenetic studies suggest that future studies are needed to examine the association of statins and CRC risk, and to assess if this is mediated by direct anti-tumor effects or by mitigating the risk from hyperlipidemia.

There is growing evidence to suggest that beta-adrenergic pathways could play an important role in cancer-mediated cell proliferation, apoptosis, and angiogenesis [127]. This has generated interest in the effects of beta-blockers on CRC risk, potentially identifying chemopreventive effects. The results from two population-based studies from the Netherlands did not suggest any potential benefits of beta-blockers compared to non-users [128,129]. However, when examining beta-blocker use by stage, patients with stage IV CRC who used beta-blockers had an improved survival rate compared to non-users [130,131]. These results were further seen in a recent meta-analysis, beta-blockers had marginally improved CRC-specific survival, but stage IV CRC patients on immunotherapy had a significantly improved progression-free survival (HR = 0.76; 95% CI: 0.62–0.92, *p* = 0.005) [132]. It is very interesting that the combination of immunotherapy and beta-blockers signals a significant improvement in survival for stage IV CRC patients. Assessing the survival benefit for the entire CRC cohort is difficult since the benefits found are either not significant or marginal. Therefore, tailored RCTs conducted for CRC patients with stage IV disease receiving immunotherapy would be of great value.

### 2.6. CRC and Supplements: Vitamin D, Calcium, and Folic Acid

The potential association between vitamin D and CRC has been extensively studied. The challenges of these studies include how vitamin D exposure is evaluated, either through dietary intake assessments with their inherent biases or through serum measurements of vitamin D in the blood. Three meta-analyses of observational studies examining plasma concentrations of vitamin D and CRC incidence found a consistent inverse association between CRC incidence and higher vitamin D concentrations [133,134,135]. Examining vitamin D intake and CRC incidence, cohort studies have also suggested an inverse association between higher vitamin D intake and lower CRC incidence [136,137]. 

CRC mortality has also been shown to have an inverse relationship with higher plasma concentrations of vitamin D and lower CRC mortality [138,139], but not with dietary intake [140]. Additionally, four RCTs have not shown an association between vitamin D supplementation and lower CRC incidence or CRC mortality [141,142,143,144]. In contrast, a recent meta-analysis of RCTs showed a 30% reduction in CRC mortality in patients with vitamin D supplementation compared to non-users (HR  =  0.70; 95% CI: 0.48–0.93), but only included 850 patients in total [145]. In addition, a meta-analysis on serum vitamin D concentration in patients with stage III CRC did not show a statistically significant association with all-cause mortality or CRC recurrence in a pooled analysis of 2628 and 2024 patients, respectively [146]. Two possible explanations might explain these contrasting results: first, there might be an underlying beneficial effect of vitamin D supplementation for a subgroup of patients who are either genetically predisposed or have susceptible tumors [147]. Secondly, the confounding effects of obesity, low physical activity, and inflammation, which are all associated with increased CRC risk, are also associated with lower serum vitamin D concentrations [148]. Thus, low vitamin D levels may reflect overall poor health rather than directly influence CRC risk.

Studies of vitamin D and CRC face numerous challenges, including evaluating exposure to vitamin D, long follow-up time to account for the time it takes for CRC to form, identifying if a subpopulation might be susceptible, and accounting for the confounding factors that affect both vitamin D concentrations and CRC risk. Future studies will have to account for these challenges to determine if vitamin D is associated with lower CRC risk. 

Observational studies have found high calcium intake to have an inverse relationship to CRC incidence [136,149] and colonic polyp reoccurrence [150,151]. However, these findings have not been replicated in more recent RCTs examining calcium supplementation, which failed to show a reduction in polyp recurrence [152]. Estimating calcium intake effects on CRC mortality, a recent study found a trend towards lower CRC-specific mortality risk in patients taking higher amounts of calcium after CRC diagnosis, but did not find any survival benefits when examining calcium intake before CRC diagnosis [153]. This lack of association with pre-diagnosis calcium intake and CRC mortality has been seen in other studies [140,154,155]. The potential protective mechanism of calcium has been proposed to be reducing proliferation and inhibiting KRAS mutations in colonic cells [156,157]. There is no current plausible explanation for why calcium supplementation should be associated with improved CRC survival post-diagnosis but not pre-diagnosis. Future studies are needed to clarify the relationship between calcium and CRC, and, in case of an association, its mechanism.

The relationship between folic acid and CRC is complicated, as studies have shown contradictory results [158]. Hubner et al. found folic acid to have a dual-modulator effect, preventive effects against CRC tumorgenesis in the absence of malignant foci, while a provoking effect in the presence of malignant foci [159]. A meta-analysis examining folic acid and CRC incidence did not find any association in RCTs or cohort studies [160]. Although folic acid has been shown to inhibit CRC cell migration in vitro [161], no association has been observed between serum folate levels and CRC survival, CRC recurrence, or overall mortality [162]. Additionally, appropriate chemotherapy for CRC includes anti-folate medications such as 5-fluorouracil and low-dose folic acid to mitigate their toxicity [163]. Collectively, there is no convincing evidence that folic acid supplements are associated with lower CRC risk. Future studies would have to take into account the dual-modulator effect in cancer that takes around 10 years to form, with measurements of supplementation use or serum folate measurements, and finally, the use of anti-folate chemotherapy.

## 3. Conclusions

Colorectal cancer is a growing challenge when considering the rising incidence rate in the world and the high mortality rate. Chemoprevention through widely used medications is an exciting and promising approach, offering cost-effective and accessible strategies even in developing countries. Low-dose aspirin has been the most extensively studied medication, with consistent, apparent beneficial effects on CRC survival, particularly among patients with COX-2 overexpression or PIK3CA mutations. Future research should aim to refine risk–benefit analyses in these genetically defined patient populations.

OAC use is increasingly hypothesized to facilitate early CRC detection via induction of GIB events, but robust studies are needed to identify hard endpoints such as tumor stage and survival outcomes. Metformin is associated with a survival benefit, most likely due to lowering of an increased risk factor in type 2 diabetes patients. Interestingly, these drugs all have either strong and consistent or emerging evidence of survival benefits for CRC patients, with very variable potential mechanisms. While corticosteroids, statins, and beta-blockers have shown mixed results, their possible roles in CRC prevention and survival warrant further exploration. Future studies conducted on chemoprevention for CRC are likely to improve and deliver meaningful clinical benefits for CRC patients, including personalized preventive strategies and optimized care.

## Figures and Tables

**Table 1 pharmaceuticals-18-00811-t001:** Overview of notable and highly cited studies conducted on aspirin and colorectal cancer (CRC) specific survival.

Observational Studies		
**Study**	**Aspirin Dose**	**CRC-Specific Survival *** **(Hazard Ratio (95% CI), *p*)** **(Relative Risk (95% CI), *p*)**
Lam et al., 2025 [6]	Low-dose **	sHR = 0.78; 95% CI: 0.76–0.81
Skriver et al., 2023 [7]	75–150 mg	HR = 0.90; 95% CI: 0.84–0.95
Shahrivar et al., 2023 [8]	75 or 160 mg	HR = 0.99; 95% CI: 0.91–1.07
Shami et al., 2022 [9]	75–300 mg	HR = 0.83; 95% CI: 0.76–0.91
Sung et al., 2019 [10]	Low-dose **	sHR = 0.69; 95% CI: 0.59–0.81
Tsoi et al., 2018 [11]	Low-dose **	sHR = 0.59: 95% CI: 0.56–0.62
Cao et al., 2016 [12]	81 or 325 mg	RR = 0.81; 95% CI: 0.75–0.88
Cook et al., 2013 [13] —8 year follow-up post-trial	100 mg	HR = 0.80; 95% CI: 0.67–0.97, *p* = 0.021
Liao et al., 2012 [14] ** PIK3CA-mutated patients	81 or 325 mg post-diagnosis	HR = 0.18; 95% CI: 0.06–0.61, *p* < 0.001
Rothwell et al., 2011 [15]	75 mg	HR = 0.60; 95% CI: 0.45–0.81, *p* = 0.0007
Rothwell et al., 2010 [16]	75 mg	HR = 0.65; 95% CI: 0.48–0.88, *p* = 0.005
Chan et al., 2009 [17]	81 or 325 mg	HR = 0.71; 95% CI: 0.53–0.95
Thun Michael et al., 1991 [18]	Low-dose **	Men: RR = 0.60; 95% CI: 0.40–0.89, *p* < 0.001 Women: RR = 0.58; 95% CI: 0.37–0.90, *p* < 0.001
**Randomized controlled studies**		
**Study**	**Aspirin Dose**	**CRC-Specific Survival *** **(Hazard Ratio (95% CI), *p*)** **(Relative Risk (95% CI), *p*)**
McNeil et al., 2018 [19]	100 mg	HR = 1.77; 95% CI: 1.02–3.06
Cook et al., 2005 [20]	100 mg	RR = 0.94; 95% CI: 0.79–1.11, *p* = 0.45
**Meta-Analysis**		
**Study**	**Aspirin Dose**	**CRC-Specific Survival *** **(Hazard Ratio (95% CI), *p*)** **(Relative Risk (95% CI), *p*)**
Mädge et al., 2022 [21]	Variable, most often low-dose **	HR = 0.74; 95% CI: 0.62–0.89
Wang et al., 2021 [22]	Variable, most often low-dose *	Cohort studies: RR = 0.85; 95% CI: 0.78–0.92 RCTs: RR = 0.74; 95% CI: 0.56–0.97
Bosetti et al., 2020 [23]	Variable, most often low-dose **	RR = 0.73; 95% CI: 0.69–0.78, *p* < 0.001
Lin et al., 2020 [24]	Variable, most often low-dose **	HR = 0.78; 95% CI: 0.73–0.85
Algra et al., 2012 [25]	Variable, most often low-dose **	OR = 0.58; 95% CI: 0.44–0.78, *p* = 0.0002
Rothwell et al., 2012 [26]	Low dose **	OR = 0.58; 95% CI: 0.38–0.89, *p* = 0.008

* HR = Hazard ratio; RR = Relative Risk; OR = Odds ratio, CI = confidence interval; sHR = Subdistribution Hazard ratio. ** Low-dose: usually between 75–100 mg, some studies did not state a specific dose but stated low-dose aspirin.

**Table 2 pharmaceuticals-18-00811-t002:** Overview of notable studies conducted on oral anticoagulation (OAC) and colorectal cancer (CRC).

Study	OAC Type	Endpoint	Results	Statistical Measurement
Rasmussen et al., 2022 [71]	DOACs Warfarin	CRC bleeding events	RR = 12.3–24.2 depending on age	Risk ratio
Abrahami et al., 2020 [72]	DOACs Warfarin	CRC incidence	DOAC: HR = 1.73; 95% CI: 1.01–2.99 Warfarin: HR = 1.14; 95% CI: 0.74–1.77	Hazard ratio
Haaland et al., 2017 [73]	Warfarin	CRC incidence	Overall group: IRR = 0.99; 95% CI: 0.93–1.06 Atrial Fibrillation subgroup: IRR = 0.71; 95% CI: 0.63–0.81	Incidence rate ratio
Clemens et al., 2014 [74]	Dabigatran Rivaroxaban Apixaban Warfarin	CRC incidence	0.20–0.52%	Cumulative incidence
O’Rouke et al., 2014 [75]	Warfarin	CRC specific mortality	HR = 0.88; 95% CI: 0.77–1.01	Hazard ratio
Johannsdottir et al., 2012 [76]	Warfarin	CRC detection via anemia screening	0.31%	Cumulative incidence

## Data Availability

No new data were created or analyzed in this study. Data sharing is not applicable to this article.

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
