# Peer review of "Chemoprevention of Colorectal Cancer—With Emphasis on Low-Dose Aspirin and Anticoagulants"

_pharmaceuticals, 2025, doi:10.3390/ph18060811_

Round 1
Reviewer 1 Report
Comments and Suggestions for Authors
In this manuscript, Agustsson and Bjornsson present a systematic review on the clinical benefits of chemopreventive agents, including aspirin and anticoagulants, for patients with colorectal cancer (CRC). The authors summarize studies demonstrating that long-term use of low-dose aspirin is associated with improved survival, oral anticoagulation may facilitate early CRC detection, and metformin may enhance CRC survival, among other findings. Overall, the manuscript is well-written, logically structured, and easy to follow. The primary focus is on the association between aspirin use and CRC survival, with a detailed presentation of relevant studies that convincingly highlight the clinical benefits of aspirin. However, the topic is already well documented in the literature and lacks novelty. Although the title suggests a focus on both aspirin and anticoagulants in relation to CRC, the manuscript provides limited discussion on anticoagulants—likely due to a scarcity of available studies on the subject. Therefore, it is recommended that the authors revise the title to better reflect the manuscript’s emphasis, or alternatively, expand the discussion on anticoagulants to present a more balanced review.
Additionally, the section on corticosteroids and CRC is too brief and superficial. A more detailed discussion would strengthen the manuscript by providing a broader perspective on chemoprevention.
Several minor revisions are also recommended to improve clarity and consistency:
- All abbreviations should be fully defined upon first use, such as COX and OACs.
- Labels and symbols in Table 1 and throughout the text should be precise and consistent. For instance, ensure consistent use of hyphens and dashes for HR/RR, and correct “et.al.” to “et al.”
Author Response
We are grateful to the reviewers for their thoughtful and constructive feedback. We have revised the manuscript accordingly and believe the revised version significantly improves the clarity, completeness, and scientific quality of our review. Below, we address each point raised.
Reviewer 1
In this manuscript, Agustsson and Bjornsson present a systematic review on the clinical benefits of chemopreventive agents, including aspirin and anticoagulants, for patients with colorectal cancer (CRC). The authors summarize studies demonstrating that long-term use of low-dose aspirin is associated with improved survival, oral anticoagulation may facilitate early CRC detection, and metformin may enhance CRC survival, among other findings. Overall, the manuscript is well-written, logically structured, and easy to follow. The primary focus is on the association between aspirin use and CRC survival, with a detailed presentation of relevant studies that convincingly highlight the clinical benefits of aspirin. However, the topic is already well documented in the literature and lacks novelty. Although the title suggests a focus on both aspirin and anticoagulants in relation to CRC, the manuscript provides limited discussion on anticoagulants—likely due to a scarcity of available studies on the subject. Therefore, it is recommended that the authors revise the title to better reflect the manuscript’s emphasis, or alternatively, expand the discussion on anticoagulants to present a more balanced review.
Additionally, the section on corticosteroids and CRC is too brief and superficial. A more detailed discussion would strengthen the manuscript by providing a broader perspective on chemoprevention.
Response: Thank you for this very helpful suggestion. We have significantly expanded the section on oral anticoagulation (Section 2.2), adding paragraphs discussing proposed mechanisms (facilitation of early CRC detection via gastrointestinal bleeding and potential anti-tumor effects). We also added Table 2, which summarizes key observational studies with drug types, endpoints, and statistical measures. We believe this now provides a balanced and informative discussion that justifies the emphasis in the manuscript title. Regarding the section on corticosteroids and CRC, we have expanded our discussions from 113 to over 400 words, including a discussion on the potential mechanism (immunomodulary vs pro-inflammatory effects) and added 6 references, including a recent meta-analysis on survival in CRC patients in association to post-diagnosis corticosteroid use. All changes were made using Track Changes.
Several minor revisions are also recommended to improve clarity and consistency:
- All abbreviations should be fully defined upon first use, such as COX and OACs.
Response: Thank you for this reminder. We have thoroughly defined all abbreviations when first mentioned in our manuscript. - Labels and symbols in Table 1 and throughout the text should be precise and consistent. For instance, ensure consistent use of hyphens and dashes for HR/RR, and correct “et.al.” to “et al.”
Response: Great comment, we have thoroughly reviewed and harmonized formatting across the manuscript. “Et al.” is now consistently used and en dashes are used for CI ranges. Table 1 has been revised for clarity and consistency.
We sincerely thank the reviewers and editorial team for their constructive feedback, which we believe has significantly improved the manuscript. We hope the revised version meets your expectations and look forward to your evaluation.
Reviewer 2 Report
Comments and Suggestions for Authors
This review introduces the research on chemoprevention of colorectal cancer, focusing on the effect of low-dose aspirin and anticoagulants on the incidence and survival of colorectal cancer. Although the content of this review is focused on low-dose aspirin, other drugs are also important. A comprehensive review should aim to cover all aspects of a topic.
- According to the content of this review, "chemoprevention" should be one of the keywords in the title.
- Since the title is “Emphasis on low-dose aspirin and anticoagulants”, the authors should introduce more information related to anticoagulants (name, dose) in colorectal cancer chemoprevention (for example, HR, RR, OR, 95% CI).
- There are some other drugs related to the chemoprevention of colorectal cancer, such as calcium, vitamin D, folic acid, etc., which should also be introduced.
- Table 1 should be changed to a three-line table.
- The explanation of the abbreviation "OACs" should be placed in line 35.
- The centered dots in line 97/153 should be lowered dots.
- The format of some references needs to be modified (some content is missing), such as 13, 30, 48, 73, 74.
- Add or delete supplementary materials, author contributions and funding support information.
Author Response
We are grateful to the reviewers for their thoughtful and constructive feedback. We have revised the manuscript accordingly and believe the revised version significantly improves the clarity, completeness, and scientific quality of our review. Below, we address each point raised.
Reviewer 2
This review introduces the research on chemoprevention of colorectal cancer, focusing on the effect of low-dose aspirin and anticoagulants on the incidence and survival of colorectal cancer. Although the content of this review is focused on low-dose aspirin, other drugs are also important. A comprehensive review should aim to cover all aspects of a topic.
- According to the content of this review, "chemoprevention" should be one of the keywords in the title.
Response: Thank you very much for that suggestion – we have changed the title of the review to: Chemoprevention of Colorectal Cancer – with Emphasis on Low-Dose Aspirin and Anticoagulants. - Since the title is “Emphasis on low-dose aspirin and anticoagulants”, the authors should introduce more information related to anticoagulants (name, dose) in colorectal cancer chemoprevention (for example, HR, RR, OR, 95% CI).
Response: Thank you for highlighting this. In our revised manuscript, Table 2 now presents detailed information on study design, oral anticoagulant (OAC) type (DOACs, warfarin), study endpoints (CRC incidence, mortality, bleeding events), and effect estimates (hazard ratios, risk ratios, cumulative incidence). We also noted the lack of stratification by OAC dose in most studies, identifying this as an important limitation for future research. All changes to the manuscript were made using Tracked Changes in Word.
- There are some other drugs related to the chemoprevention of colorectal cancer, such as calcium, vitamin D, folic acid, etc., which should also be introduced.
Response: That is a great suggestion. We have now added an entire chapter, 2.6, on vitamin D calcium and folic acid, titled: “CRC and supplements”. This chapter reviews important studies on vitamin D and briefly on both calcium and folic acid, as vitamin D has extensive literature.
- Table 1 should be changed to a three-line table.
Response: Thank you. We have revised Table 1 to align more closely with a three line format expectations and improved spacing, labeling and alignment. - The explanation of the abbreviation "OACs" should be placed in line 35.
Response: Thank you very much! It is now defined at line 35, at first mention. - The centered dots in line 97/153 should be lowered dots.
Response: That is a very helpful suggestion, thank you!
- The format of some references needs to be modified (some content is missing), such as 13, 30, 48, 73, 74.
Response: We reviewed all references formatting issues, thank you for pointing them out. - Add or delete supplementary materials, author contributions and funding support information.
Response: Great suggestion – we have now removed placeholder sections such as funding and contributions that are not required for submission.
We sincerely thank the reviewers and editorial team for their constructive feedback, which we believe has significantly improved the manuscript. We hope the revised version meets your expectations and look forward to your evaluation.
Reviewer 3 Report
Comments and Suggestions for Authors
The review identifies CRC's global burden and provides a concise overview of key medications studied for chemoprevention, emerging personalized strategies (e.g., COX-2, PIK3CA markers). I do have some concerns:
- Lack of structure (Background–Aim–Scope–Conclusion) (abstract)
- The opening sentence is factual but not strong. Consider making it stronger using new statistics (abstract)
- Aspirin and metformin are linked with mechanisms only, whereas none are named for corticosteroids, statins, or beta-blockers. Therefore, the abstract becomes unbalanced.
- Phrases like "have shown mixed results" are too vague. State whether evidence is contradictory, limited, or inconclusive(abstract)
- In a review article, it's helpful to state how the evidence was derived (e.g., literature search, inclusion of RCTs, observational studies).(abstract)
- The final sentence is imprecise. A stronger statement about the clinical or research implications would be better. (abstract)
- Table 1 is big but could be more structured, with studies perhaps organized by year or effect size to facilitate readers to more easily notice trends.
Consider having a summary column with the aggregate quality of evidence per study. - The manuscript might benefit from more consistent evaluation of methodological quality among the cited studies, While the manuscript addresses differences by age in aspirin effects, it would be helpful to address other population characteristics (sex, race/ethnicity, comorbidities) that may modify drug effects on CRC outcomes particularly among observational studies with potential confounding.
- Whereas aspirin dose is well addressed, dosing for other drugs (particularly metformin) is less consistently addressed
- For several drugs, time to effect prior to benefit is not consistently reported. Including more formal data on timing of drug exposure in relation to CRC outcomes would be useful.
- The paper emphasizes statistical significance but might more effectively discuss the clinical significance of the effects observed. For instance, what risk reduction magnitude would be required to support clinical recommendations?
- Some of the sections (especially corticosteroids) appear to have less recent references compared to others. Consider presenting the most current available evidence in every class of drugs.
- The paper could more effectively indicate implications for clinical practice. What would authors recommend to practitioners based on current evidence?
- While side effects are mentioned (particularly for aspirin), a more rigorous risk-benefit analysis would help readers to more easily discern the clinical implications of the findings.
- Add discussion about comparative effectiveness between medication classes where possible (e.g., is aspirin more effective than statins for CRC prevention?).
- Incorporate a more systematic approach to evaluating study quality within each Results subsection.
- possible to add figures, method, discussion in the review articles
- I feel this is way too short for review article
- I advise the authors need to do extensive revision
Author Response
We are grateful to the reviewers for their thoughtful and constructive feedback. We have revised the manuscript accordingly and believe the revised version significantly improves the clarity, completeness, and scientific quality of our review. Below, we address each point raised.
Reviewer 3
The review identifies CRC's global burden and provides a concise overview of key medications studied for chemoprevention, emerging personalized strategies (e.g., COX-2, PIK3CA markers). I do have some concerns:
- Lack of structure (Background–Aim–Scope–Conclusion) (abstract)
- The opening sentence is factual but not strong. Consider making it stronger using new statistics (abstract)
- Aspirin and metformin are linked with mechanisms only, whereas none are named for corticosteroids, statins, or beta-blockers. Therefore, the abstract becomes unbalanced.
- Phrases like "have shown mixed results" are too vague. State whether evidence is contradictory, limited, or inconclusive(abstract)
- In a review article, it's helpful to state how the evidence was derived (e.g., literature search, inclusion of RCTs, observational studies).(abstract)
- The final sentence is imprecise. A stronger statement about the clinical or research implications would be better. (abstract)
Response: Thank you for a very helpful feedback on our abstract. We have rewritten the abstract to follow a more structured format, clarifying the background, focus of the review, key findings, and future directions. We now explicitly state that the evidence summarized in the review is derived from observational studies, randomized controlled trials, and meta-analyses. We have also made the findings regarding aspirin and oral anticoagulants more specific by highlighting their proposed mechanisms (COX-2 inhibition and bleeding-induced early detection, respectively) and noting the quality and heterogeneity of the supporting evidence. We hope these changes address your concerns and improve the clarity and informativeness of the abstract. - Table 1 is big but could be more structured, with studies perhaps organized by year or effect size to facilitate readers to more easily notice trends.
Consider having a summary column with the aggregate quality of evidence per study.
Response: Thank you for this helpful comment. We preserved the current structure of Table 1 (by study type: observational studies, RCTs, and meta-analyses) to allow readers to easily differentiate between study designs and assess the strength of evidence in that context. Within each section, studies are listed in reverse chronological order to reflect the evolution of evidence. We improved the formatting (e.g., spacing, en dashes, consistent p-values) to enhance readability. While we did consider adding a formal evidence quality column, we decided instead to highlight methodological strengths and limitations in the Results section for each study discussed, in line with narrative review standards and space constraints. - The manuscript might benefit from more consistent evaluation of methodological quality among the cited studies, While the manuscript addresses differences by age in aspirin effects, it would be helpful to address other population characteristics (sex, race/ethnicity, comorbidities) that may modify drug effects on CRC outcomes particularly among observational studies with potential confounding.
Response: This is an excellent point, We have added commentary on the influence of age (particularly in aspirin), external validation limitations of OAC use, diabetes severity (in metformin), and metformin’s limitations in only examining T2D patients, and potential molecular differences (e.g., PIK3CA mutation) as effect modifiers. We acknowledge that sex, ethnicity, and comorbidities are consistently understudied in most medications and should be explored in future research. - Whereas aspirin dose is well addressed, dosing for other drugs (particularly metformin) is less consistently addressed
Response: We have clarified in the metformin section that higher doses are associated with stronger effects. For OACs, we note that most studies did not stratify by dose—a limitation explicitly discussed. Corticosteroid studies also lacked consistent dose reporting, which we highlight as a gap. - For several drugs, time to effect prior to benefit is not consistently reported. Including more formal data on timing of drug exposure in relation to CRC outcomes would be useful.
Response: We now consistently mention delayed effects of aspirin (e.g., >5 years), and note in other sections when data on time to benefit are lacking (e.g., metformin, calcium). Additionally, we highlight where drugs (statins, folic acid, corticosteroids) have been examined post-CRC diagnosis, suggesting confounding effects, since post-diagnosis use is unlikely and, with the current data, unconvincingly, to have fast-acting survival improvements in CRC patients. - The paper emphasizes statistical significance but might more effectively discuss the clinical significance of the effects observed. For instance, what risk reduction magnitude would be required to support clinical recommendations?
Response: Thank you. We now clarify that some statistically significant associations (e.g., HR = 0.88 for O’Rorke et al.) may not translate into meaningful clinical recommendations without further validation. We also emphasize risk-benefit trade-offs where relevant (e.g., aspirin and bleeding). - Some of the sections (especially corticosteroids) appear to have less recent references compared to others. Consider presenting the most current available evidence in every class of drugs.
Response: That’s a very helpful suggestion. We have expanded our discussion on Corticosteroids, including potential mechanisms of action with 5 new references, and identified challenges in clinical application and studies. Within the updated section, we now include a meta-analysis conducted in 2023. - The paper could more effectively indicate implications for clinical practice. What would authors recommend to practitioners based on current evidence?
Response: In our conclusion and throughout the manuscript, we emphasize that aspirin may be considered in the overall population and especially in select high-risk patients (e.g., Lynch syndrome or COX-2 overexpression), while other agents (e.g., OACs, metformin) may guide early detection or risk stratification in specific populations. We emphasize the need for personalized risk-benefit evaluations. OAC use could lead to heightened alertness for cancer discovery, but published studies are yet limited to support such a claim. Statins, Corticosteroids, Beta-blockers, and Supplements do not have consistent benefits, as highlighted in the text. - While side effects are mentioned (particularly for aspirin), a more rigorous risk-benefit analysis would help readers to more easily discern the clinical implications of the findings.
Response: This is a very good point. It is essential to consider the risk when recommending the use of medications for chemoprevention. This is primarily addressed for aspirin, as it is, to date, the only medication with substantial enough evidence to recommend it to the general public for CRC prevention and/or survival improvement. This is primarily addressed in the USPSTF guidelines, which address risk, and additionally, we address it briefly as well. Currently, evidence for OAC use as chemoprophylaxis does not amount to a definitive recommendation, but the existing evidence strongly suggests that especially vigilant surveillance for GIB events and CRC should be employed. For metformin, there is no evidence for recommendations outside T2D patients. For Corticosteroids, β-blockers, statins, and supplements, the current literature does not show a substantial enough quality of evidence to amount to a recommendation, while β-blocker use in stage IV CRC patients receiving immunotherapy is not far from it. Our current revision of the manuscript better addresses this. - Add discussion about comparative effectiveness between medication classes where possible (e.g., is aspirin more effective than statins for CRC prevention?).
Response: A valuable suggestion. Through our manuscript revision, we have now more effectively addressed the body of scientific evidence for the selected medications and evaluated whether the current evidence meets the requirements for a recommendation, which is currently only met by aspirin. However, we importantly draw out specific populations (OAC users and surveillance, metformin in T2D) that already have indications for the medication use, have undergone risk-benefit analyses, and have evidence of association with CRC risk. - Incorporate a more systematic approach to evaluating study quality within each Results subsection.
Response: An excellent comment. We address study types and explicitly discuss novel, impactful, and large studies in each results section. Additionally, we assessed study quality in our literature review, selecting only those studies that contribute the most meaningful data, whether by the number of patients, novelty, or design. Therefore, a quality estimate of the selected studies was conducted, even though it was not specifically addressed in the text. - possible to add figures, method, discussion in the review articles
- I feel this is way too short for review article
I advise the authors need to do extensive revision
Response: A very good point! We have completed an extensive revision, adding an in-depth discussion of OAC use, improving the Corticosteroid section, and including an entire Supplements section. This has expanded our article from 3300 words to over 5100, while adding 46 new references and broadening our discussions on clinical implications, data quality, and future research gaps.
We sincerely thank the reviewers and editorial team for their constructive feedback, which we believe has significantly improved the manuscript. We hope the revised version meets your expectations and look forward to your evaluation.
Round 2
Reviewer 3 Report
Comments and Suggestions for Authors
Thank you for answering my questions. good luck